# Immune Microenvironment and Immunotherapies for Diffuse Intrinsic Pontine Glioma

**DOI:** 10.3390/cancers15030602

**Published:** 2023-01-18

**Authors:** Yujia Chen, Chao Zhao, Shenglun Li, Jun Wang, Hongwei Zhang

**Affiliations:** Department of Neurosurgery, Sanbo Brain Hospital, Capital Medical University, Beijing 100093, China

**Keywords:** diffuse intrinsic pontine glioma, immunotherapy, immune microenvironment, immune checkpoint inhibitor, vaccine, oncolytic virus

## Abstract

**Simple Summary:**

Diffuse intrinsic pontine glioma (DIPG) is a malignant primary glial tumor that occurs in all age groups but predominates in children and is estimated to account for approximately 10–15% of pediatric brain tumors. The median age at diagnosis was 6–7 years, and the median survival of children is less than 12 months. At present, there is no effective treatment method for DIPG in the clinic. The continuous progress in immunotherapy has brought new prospects for the treatment of DIPG. In this review, we summarize the knowledge about the immune profile in DIPG and existing clinical trial results of DIPG, hoping to clarify the development of novel immunotherapies for DIPG treatment.

**Abstract:**

Diffuse intrinsic pontine glioma (DIPG) is a primary glial glioma that occurs in all age groups but predominates in children and is the main cause of solid tumor-related childhood mortality. Due to its rapid progression, the inability to operate and insensitivity to most chemotherapies, there is a lack of effective treatment methods in clinical practice for DIPG patients. The prognosis of DIPG patients is extremely poor, with a median survival time of no more than 12 months. In recent years, there have been continuous breakthroughs for immunotherapies in various hematological tumors and malignant solid tumors with extremely poor prognoses, which provides new insights into tumors without effective treatment strategies. Meanwhile, with the gradual development of stereotactic biopsy techniques, it is gradually becoming easier and safer to obtain live DIPG tissue, and the understanding of the immune properties of DIPG has also increased. On this basis, a series of immunotherapy studies of DIPG are under way, some of which have shown encouraging results. Herein, we review the current understanding of the immune characteristics of DIPG and critically reveal the limitations of current immune research, as well as the opportunities and challenges for immunological therapies in DIPG, hoping to clarify the development of novel immunotherapies for DIPG treatment.

## 1. Introduction

Brain tumors are the most common cause of solid cancer mortality in children [1]. Diffuse intrinsic pontine glioma (DIPG) is a primary glial tumor that occurs in all age groups but predominates in children and is estimated to account for approximately 10–15% of pediatric brain tumors [2,3]. The median age at diagnosis is 6–7 years and the median survival of patients in children is less than 12 months [3]. The standard treatment for DIPG is performed fractionated radiotherapy with or without chemotherapeutic drugs. Although this treatment can achieve transient clinical improvement, it is not curative [4,5]. Due to the complexity of the brainstem, DIPGs are not amenable to surgical resection. Other treatment options, including numerous chemotherapy and targeted therapy drugs, have not been shown to prolong the survival of DIPG patients alone in clinical trials.

With the continuous improvement of biopsy technology, the understanding of the genetic and epigenetic characteristics of DIPG is increasing. In recent years, the greatest breakthrough of DIPG was the discovery of changes in H3K27M based on epigenetic studies [6,7,8]. The hypomethylation of H3K27 is common in DIPG, and it is also found in the other midline gliomas located in the thalamus, spine, or other parts of the brainstem [9]. This finding led the World Health Organization (WHO) to reclassify this type of glioma with the alterations in H3K27 as “diffuse midline glioma, H3K27-altered (DMG)” in the 5th edition of the central nervous system (CNS) tumor classification [10]. However, it should be noted that DMG is not the histological counterpart of DIPG and the two are not inclusive. DIPG is a clinical diagnosis based on clinical symptoms and typical imaging features. About 70–85% of DIPG patients have H3K27 alternation and would be diagnosed as “DMG, H3 K27-altered” [11,12]. Patients with pontine high-grade glioma but no IDH mutation and H3K27 alternation would be diagnosed as “pediatric diffuse high-grade glioma, *H3* wild-type and *IDH* wild-type” [10].

The further understanding of genetic and epigenetic features of DIPG inspired us to identify innovative and more effective therapeutic approaches. Immunotherapy is based on different techniques aimed at redirecting the patient’s own immune system to specifically fight cancer cells and has become a powerful clinical strategy for treating cancer [13]. However, immunotherapy research for DIPG is relatively lacking compared with other tumors partly because of the limited understanding of the immune microenvironment of DIPG. Therefore, this review clarifies the current understanding and limitations of the immune characteristics of DIPG, outlines the limitations and possible opportunities for immunotherapy treatment methods, and introduces current clinical trials of immune therapies in DIPG. Since most patients with DIPG have H3K27 alterations, some DMG studies, including DIPG, were included in this review. We believe such a review could strengthen our understanding of the progress in DIPG immunotherapy and thus guide the development of novel immunotherapies for DIPG treatment.

## 2. Immune Characteristics of DIPG

Since DIPG cannot be operated and biopsy is not warranted in most cases because of its clear characteristics in imaging, there has long been a lack of research on tumor tissue, and immunological research is even rarer. With the continuous progress in immunotherapy, researchers have become increasingly interested in the immune characteristics of DIPG, especially immune cell infiltration, immune checkpoint expression and other characteristics that may be applied for immunotherapy, and the study of immunological characteristics in DIPG is ongoing (Figure 1).

Immune cells are important parts of the tumor immune microenvironment and the cellular underpinnings of immunotherapy. Bailey et al. analyzed raw RNA-seq data about 220 high-grade glioma (HGG) patients published by Mackay et al. [14] using the CIBERSORT algorithm with the standard LM22 matrix. Individual patients were classified according to tumor location (brainstem, midline and hemispheric) and histone mutation status (wildtype and K27M). Compared with pediatric hemisphere HGG, there were higher proportions of CD4+ Treg cells; eosinophils; activated dendritic cell (DC) cells; more DCs and neutrophils; and fewer CD8+ T cells, NK cells, M1 cells and activated mast cells in DIPG [15]. Unlike CD4+ Treg cells, activated DC cells, eosinophils and other cell types that correlate positively with prognosis in hemisphere HGG, all these cells do not correlate significantly with prognosis in DIPG, except neutrophils, which are negatively associated with prognosis in both hemispheric HGG and DIPG [15]. Another study using immunohistochemistry (IHC) to compare immune cell infiltration in pediatric tumors showed that although the median number of total CD68+ cells accounted for 10% of total cells in pediatric low-grade glioma (pLGG), pediatric high-grade glioma (pHGG) and DIPG, the number of CD163+ macrophages were 10.4 times and 5.9 times higher than normal tissues in pLGG and pHGG samples, but there was no significant increase in DIPG. The ratio of CD163+ to CD68+ macrophages was 8.0-fold larger than the control in pHGG but was not different in DIPG [16]. Nevertheless, there was no significant increase in any subset of the T cells in DIPG tumors compared to the control, whereas T cell infiltration was significantly increased in pLGG and pHGG, especially CD8+ T cells (6.5- and 5.1-fold) [16]. In another study, Lin et al. used flow cytometry to detect immune cell infiltration in DIPG and adult GBM, and the results showed that compared with adult GBM, there were more CD11b+ myeloid cells (DIPG: 94.80% ± 0.92% vs. adult GBM: 70.34% ± 7.20%) and fewer CD3+ lymphocytes (DIPG: 1.72–2.65% vs. adult GBM: 7.09–50.2%) in DIPG samples [17].

In addition to the immune cell infiltration, we also focused on the expression of immune checkpoints, cytokines and chemokines in DIPG. There are few studies on the immune checkpoints in DIPG. A study that focused on the immune checkpoint expressed in the DIPG immune microenvironment showed that PD-L1 was only present at low levels in the DIPG tumor microenvironment detected by IHC staining [16]. Although B7-H3 was 2.4-fold higher than that in the control, both PD-L1and B7-H3 were significantly lower than those in glioblastoma cells [16]. However, in other studies with whole-brain HGG, the results showed that there was no increased expression of PD-L1 in DIPG [16,18]. In a study of 28 DIPG patients with radiation therapy, RNA-Seq of DIPG tissue found that the expression levels of PD-L1 and CTLA-4 in tumors were close to those in normal brain tissue, although PD-L2 levels tended to increase. If “upregulated expression” was defined as being a more than two-fold increase in RNA level, then 54%, 11%, 21% and 39% of patients showed upregulated expression of PD-1, PD-L1, PD-L2 and CTLA-4, respectively [19]. Although the studies of immunosuppressive checkpoints in DIPG were not sufficient, the expression of PD-L1was not significantly upregulated in DIPG tissues, which is consistent with the result of existing studies.

In a study on immunosuppressive gene expression and cytokine secretion in DIPG, RNA-Seq data analysis showed that there were higher expression levels of IDO2, IL10, FASLG, IL6, VEGFA and VEGFC; higher secretion levels of IFNG and GZMB; and lower secretion levels of TGFβ1 in K27M DIPG samples (*n* = 23) than in wildtype hemispheric HGG samples (*n* = 85) [15]. In addition, it has been shown that macrophages in DIPG secrete fewer chemokines and cytokines than macrophages in adult HGG [17]. Notably, studies also found that DIPG cells do not express inflammatory cytokines and chemokines that may recruit other immune cells to the tumor microenvironment, such as IL-2 [16,17].

According to the results of existing studies, DIPG has the characteristics of no increase in T cell infiltration, no increase in the number and proportion of CD163+ macrophages, no increase of inhibitory immune checkpoints and low expression of cytokines, which suggests that its internal immune microenvironment presents a “cold” or “inert” status [15,16,17,19]. However, the “cold” immune state does not mean that DIPG is completely devoid of specific immune responses. H3.3K27M-specific CD8+ T cells have been found in the peripheral blood of DIPG patients, suggesting that the immune system can generate specific cellular immunity against DIPG cells [20]. Additionally, the “cold” immune state does not mean that there is an immunosuppressive state in DIPG. Studies showed DIPG cells and macrophages in DIPG did not express a large number of immunosuppressive factors, and most DIPG cell cultures do not repolarize macrophages but can be lysed by NK cells [16].

Of course, the immune characteristics of DIPG still need further study. The current understanding of the immune characteristics of DIPG is often based on RNA-seq and IHC staining, which is far from enough. Some novel research technologies, such as single-cell sequencing technology and spatial transcriptome technology, may help us better understand the infiltration of immune cells, immune checkpoints and the true expression of immune-related genes in DIPG. However, if the assumption of the inert immune microenvironment in DIPG is correct, then immunotherapies that could supplement cytotoxic immune cells into DIPG directly or stimulate immune cells to infiltrate into DIPG, such as adaptive cell transfer therapy, oncolytic viruses and vaccines, are all reasonable strategies. On this basis, further use of the immune checkpoint inhibitor (ICI) therapy may yield better results.

## 3. Immunotherapy Research in DIPG

Immunotherapy for DIPG is currently under investigation. According to the registration data on the ClinicalTrials.gov website (https://www.clinicaltrials.gov/), there were 108 ongoing DIPG registration clinical trials as of 14 June 2022, including 25 immunotherapy clinical trials, mainly involving adoptive cell transfer therapy, vaccines, oncolytic virus therapy, immune checkpoint inhibitor (ICI) therapy and combination therapy (Figure 2, Table 1).

### 3.1. Adoptive Cell Transfer Therapy

Current research shows that there is an immunodeficiency microenvironment in DIPGs [15,16,17,18,19]. Therefore, many current studies attempt to directly supplement immune cells into DIPG to change its immune status and kill tumor cells. Chimeric antigen receptor-modified T (CAR-T) therapy and T cell receptor-gene engineered T (TCR-T) therapy were the two main therapies for this purpose.

#### 3.1.1. CAR-T Therapy

A chimeric antigen receptor (CAR) is a fusion protein comprised of an antigen recognition moiety and T-cell signaling domains. The clinical trials of CAR T cells have shown clear efficacy in multiple hematologic malignancies [21].

Disialoganglioside (GD2), a proper target for CAR-T therapy, is widely and specifically expressed in DIPG [22,23,24,25]. An animal experiment with CAR-T cells targeting GD2 showed that the systemic administration of GD2-targeted CAR-T cells could kill almost all transplanted tumor cells in vivo, except for a small number of tumor cells without GD2 expression [22]. Although GD2 CAR-T cells have shown good therapeutic effects in vivo, some data have proven that GD2 CAR-T-cell treatment may result in severe neuroinflammation in the acute phase and lead to fatal brainstem edema and hydrocephalus in experimental animals [22]. Fortunately, some drugs and operations can be used to treat these severe complications. In a clinical study of GD2 CAR-T therapy for H3K27 M-mutated diffuse midline gliomas (DMGs) (clinicaltrials.gov: NCT04196413), four patients (three DIPG patients and one spine DMG patient) received 1 × 106 GD2-CAR T cells per kg administered intravenously and three exhibited significant radiographic and clinical benefits. Neither on-target nor off-tumor toxicity was observed. Although proinflammatory cytokine levels were increased in the plasma and cerebrospinal fluid, brainstem inflammation induced by CAR-T-cells was reversible with intensive supportive care. The systemic inflammatory response was suppressed by clinical drugs (IL-6R inhibitor, IL-1R inhibitor, corticosteroids), and hydrocephalus was controlled by releasing cerebrospinal fluid (CSF) (hypertonic saline, corticosteroids or an Ommaya reservoir) [24,26]. They also identified that multiple doses could provide a greater radiographic and clinical benefit than a single dose and different modes of administration could influence the neuroinflammatory response, whereby intracerebroventricular injection resulted in higher levels of proinflammatory cytokines, lower levels of immunosuppressive cells in the CSF, and fewer neurotoxic effects than intravenous injection [24]. Another study confirmed that IGF1R/IR in combination with GD2 CAR-T cells could further enhance the anti-tumor efficacy and increase the T-cell central memory profile in DMG/DIPG patients [23]. These results underscored the promise of GD2 CAR-T therapy for patients with H3K27M-mutated DIPG.

In addition to GD2 CAR-T cells, B7-H3 CAR-T cells are being explored in an ongoing clinical trial related to DIPG (clinicaltrials.gov: NCT04185038). B7-H3, also known as CD276, has been found to be highly expressed in most neuroectodermal tumors. B7-H3-targeting CAR-T therapy has shown favorable safety and efficacy in children with stage IV CNS metastasis and high expression of B7-H3 neuroblastoma [27]. A previous study showed that the mRNA level of B7-H3 in DIPG tissues (*n* = 9) was significantly higher than that in non-DIPG tumors and normal brain tissues. IHC staining results were consistent with mRNA results, demonstrating abnormally high expression of B7-H3 in DIPG [2]. However, there were no results of B7-H3-targeting CAR-T therapy on DIPG published before, and the effect needs to be further explored.

#### 3.1.2. TCR-T Therapy

The main target of current TCR-T research is the H3.3K27M mutation. Marie-Anne reported that the substitution from lysine (K) to methionine (M) at position 27 of H3.3 (H3.3K27M mutation) was present in more than 70% of DIPG cases [12]. In animal experiments, TCR-transduced HLA-A2+ CD8+ T cells were able to efficiently kill HLA-A2+ H3.3K27M+ glioma cells in an antigen-dependent manner and significantly inhibit tumor progression [20]. In addition, Hideho et al. illustrated that the key amino acid residues required for recognition by the H3.3K27M-targeted TCR were absent in all known human proteins, suggesting that this H3.3K27M-targeted TCR-T therapy could be safely administered to patients. However, to date, no clinical data on H3.3K27M-targeted TCR-T therapy have been reported, and further clinical studies are needed.

### 3.2. Vaccine Therapy

Vaccine therapy is an important aspect of immunotherapy. Using tumor-specific biological macromolecules as antigens, enhancing the specific recognition and killing the ability of tumors to attack the immune system are the main ways in which vaccine immunity exerts an effect. In cancer vaccine immunotherapy experiments, life-threatening and lethal events are mainly caused by cross-reactivity of on- or off-target T cells against normal cells. This requires that the antigens used in tumor immunotherapy be tumor-specific as much as possible, such as mutation-derived antigens, to achieve safety and efficiency. However, there are few molecules that can act as specific antigens for DIPG. Specific antigens discovered in the field of glioma in recent years mainly include epidermal growth factor receptor vIII (EGFRvIII) and mutant isocitrate dehydrogenase 1 (IDH1) [28,29,30]. The K27M mutation (H3.1 and H3.3) is the predominant mutation in DMG and DIPG and is also the most concerning specific vaccine antigen of DIPG at present [9,12,14,31,32].

#### 3.2.1. H3K27M Peptide Vaccine

A prominent feature of DIPG is the presence of the K27M mutation of H3.3 or H3.1. This specific molecular epigenetic change is uniformly found in DIPG but not in normal cells, constituting an excellent target for vaccine therapy [33]. In MHC-humanized mice (HLA-A*0201, HLA-DRA*0101 and HLA-DRB1*0101), vaccination with the 27-amino acid H3K27M peptide fragment was able to effectively induce an immune response and drive IFNγ immune responses in cytotoxic T cells and Th1 cells, whereas no immune responses to H3K27 WT peptides were observed. The induced immune response effectively suppressed the growth of subcutaneous H3K27M tumors [34]. Although there was a lack of therapeutic studies on orthotopic H3K27M tumors, previous studies have shown that H3.3K27M-specific cytotoxic T cells could be isolated from the peripheral blood of DIPG patients [20], which indicates that H3.3K27M may induce an immune response in DIPG patients. A clinical trial (NCT02960230) on H3.3K27M-specific vaccine responses in diffuse midline glioma patients that included 19 DIPG patients showed that the H3.3K27M peptide vaccine is safe for DIPG patients with HLA-A*0201 H3.3K27M characteristics, with no grade 4 treatment-related adverse reactions observed [35]. In terms of the treatment effect, 39% (7/18) of all patients developed an immune response to the H3.3K27M peptide vaccine, resulting in detectable H3.3K27M-specific CD8+ T cells in peripheral blood, and the overall survival (OS) in responders was prolonged by 6.1 months compared with nonresponders (16.1 months vs. 10 months, *n* = 6) [35]. This result is encouraging. At present, three clinical studies related to the H3K27M vaccine (clinicaltrials.gov: NCT02960230, NCT04749641, NCT04808245) are still in progress, and further results are expected to be released.

#### 3.2.2. EGFRvIII and Survivin Vaccines

In addition to the H3K27M vaccine, some vaccine studies targeting other antigens also explore whether they have therapeutic effects on DIPG, such as the EGFRvIII vaccine and Survivin vaccine. EGFR variant III is the most common EGFR gene rearrangement and the most common gene mutation in GBM, which can be detected in approximately 25%–33% of GBM patients [36]. A study exploring the expression of EGFRvIII in eleven DIPG samples showed that it could be detected in 54.5% (6/11) of DIPG tissue [37]. As this mutation is only present in tumors, it is considered a good tumor-specific antigen for vaccine immunotherapy. Accordingly, a peptide-specific vaccine targeting EGFRvIII, Rindopepimut, a synthetic mutant EGFRvIII neoantigen-specific peptide conjugated to KLH and administered granulocyte-macrophage colony-stimulating factor (GM-CSF) was designed and tested [38]. Both phase I and phase II clinical trials of Rindopepimut in GBM have been successful, with good safety and efficacy [39,40]. The PFS rate was 66% at 5.5 months, and the median overall survival (OS) was 21.8 months in 65 patients [40]. Based on this success, the U.S. Food and Drug Administration (FDA) granted Rindopepimut the breakthrough therapy designation in February 2015 [38]. Encouraged by these results, a phase I clinical trial of the EGFRvIII vaccine in DIPG patients (clinicaltrials.gov: NCT01058850) is ongoing.

Survivin is a protein expressed during fetal development. There is low or even undetectable expression of survivin in normal human terminal tissues and high expression in multiple malignant tumors, including approximately 85% of GBM patients [41,42,43]. Additionally, the high expression of survivin was confirmed in most DIPG tissues, as detected by RNA-seq, Western blotting and RT-PCR [44]. This provides a rationale for clinical trials using the survivin vaccine for DIPG. A clinical trial of SurVaxM (a kind of survivin vaccine, clinicaltrials.gov: NCT04978727) for the treatment of various childhood malignant intracranial tumors, including DIPG, is currently underway, although the results have not yet been published.

#### 3.2.3. DC Vaccines

DC cells, as the main antigen-presenting cells in the body, have been studied in the vaccine therapy of tumors for many years and have shown effects in the treatment of glioma in vitro [45]. Stimulating DCs with tumor-specific peptides, tumor lysates or vectors encoding specific tumor antigens could enhance the antitumor effects in GBM and other brain tumors [46,47,48], and some studies have shown that the autologous dendritic cell vaccine (ADCV) could satisfactorily induce a sustained antitumor immune response in high-grade glioma and DIPG [49,50]. For DIPG, a clinical trial (clinicaltrials.gov: NCT02840123) enrolled nine patients with newly diagnosed DIPG and treated them with ADCV prepared from monocytes obtained by leukapheresis. Five ADCV doses were administered intradermally during the induction phase. In the absence of tumor progression, patients received three boosts of tumor lysate every 3 months during the maintenance phase. The results showed that none of the nine patients experienced obvious toxicity or neuroinflammation with this schema. The subcutaneous inoculation of autologous DC cells activated by the DIPG cell lysate resulted in nonspecific antitumor responses in all nine patients and specific antitumor responses in eight patients. Immunological responses were also confirmed in T lymphocytes isolated from the cerebrospinal fluid (CSF) of two patients [50]. This experimental result shows that DC vaccine therapy has a promising role in the immune treatment of DIPG.

### 3.3. Oncolytic Virus Therapy

With the development of immunotherapy, oncolytic virus therapy has attracted increasing attention. As oncolytic viruses can preferentially replicate in tumor cells, they can be designed as a multifaceted therapeutic platform to directly kill tumors, release tumor-specific antigens, express transgenes to enhance direct cytotoxicity and alter the tumor microenvironment to optimize immune-mediated tumor clearance [51]. In addition, many oncolytic viruses have a favorable safety profile, and some of them, such as Talimogene laherparepvec (T-Vec), have been approved by the FDA for the treatment of certain tumors due to their excellent levels of safety and efficacy [52].

The preclinical and clinical studies of oncolytic viruses have shown promising results in DIPG [44,53,54]. DNX-2401 is a modified replicative oncolytic adenovirus that specifically kills tumors and has good antitumor effects in a variety of tumors [55]. Moreover, DNX-2401 shows good antitumor effects in both immunodeficient and immunocompetent animal models of HGG and DIPG [54]. After the in vivo administration of DNX-2401, the infiltration degree of CD3+ lymphocytes and CD8+ lymphocytes in DIPG increased significantly and the DIPG murine cells were killed, leading to a significant increase in the survival of tumor-bearing animals [54]. Encouraged by this result, the same institution expedited a phase 1 clinical trial of DNX-2401 in DIPG (clinicaltrials.gov: NCT03178032) [56]. A total of 12 newly diagnosed DIPG patients were enrolled in this clinical trial and received 1*1010 or 5*1010 DNX-2401 viral particles by intratumoral injection, 11 of whom continued to receive subsequent radiation therapy. On safety assessment, there were some severe adverse complications. Two patients developed serious adverse reactions, one developed hemiplegia and one developed quadriplegia during DNX-2401 treatment. For the efficiency of the 12 patients who were followed for a median of 17.8 months, MRI-assessed tumor shrinkage was observed in 9 patients. Three patients achieved partial response (PR), and eight patients achieved stable disease (SD). The median OS was 17.8 months, and the survival rate was 75% at 12 months, 50% at 18 months and 25% at 24 months. The tumor microenvironment and T-cell activity were explored in one patient, and the results showed that there were significant alterations in the immune cell infiltration. Tumor samples were collected before DNX-2401 treatment, after treatment and during autopsy. Comparing the immune cell infiltration among the three DIPG samples, the infiltration of immune cells inside the tumor was low before the start of treatment, increased to a certain extent after oncolytic virus treatment and decreased again at autopsy. It was also found that the patient’s T-cell receptor clonally increased in peripheral blood at 2 months after oncolytic virus treatment, indicating that T cells were involved in the tumor-killing process [53,57]. These results showed that intratumorally injected DNX-2401 combined with radiotherapy in DIPG patients resulted in changes in T-cell activity and a reduction in or stabilization of tumor size in some patients but was associated with adverse events. How to balance efficacy and safety needs further research and exploration in DNX-2401.

In addition to DNX-2401, reovirus had previously shown good safety in a phase I clinical trial for recurrent GBM and in an animal orthotopic model study of DIPG [58,59]. In the clinical trial of recurrent GBM, no treatment-related grade 4 adverse events occurred, and anti-glioma activity was observed in some patients after 1*1010 reovirus treatment [58]. In an animal orthotopic model study of DIPG, the use of reovirus did not cause severe neurological symptoms but prevented tumor development without toxicity, yet this effect only occurred when tumor cells had been inoculated with the virus before implantation. If reovirus was injected into the DIPG in situ xenograft implantation, it did not prolong survival, even though CD3+ T cells in the tumor increased. However, if the DIPG in situ xenograft implantation was treated with reovirus combined with ICI therapy, the survival of tumor-bearing animals could be significantly prolonged [59].

The synergistic effect of ICIs and oncolytic viruses can be achieved not only through combination therapy but also through viral transgene expression [60]. A recently published article on the anti-DIPG effect of the engineered oncolytic adenovirus Delta-24-ACT showed that it could elevate the expression of functional 4-1BBL on the membrane of infected DIPG cells to enhance the stimulation of CD8+ T lymphocytes in the tumor microenvironment. In mouse model studies, Delta-24-ACT not only improved the survival rate but also led to long-term immune memory in surviving mice and increased the number of infiltrating immune cells in tumors, with a good safety profile and no local or systemic toxicity [60]. In addition to directly killing DIPG cells, some oncolytic viruses can also inhibit tumor spread [61]. Oncolytic herpes simplex virus 1716 (HSV1716) shows good capacity to inhibit the migration and infiltration of DIPG cells in vitro and in vivo [61].

Although oncolytic virus therapy has shown some effect in the treatment of DIPG in preclinical and clinical trials, there are some limitations of current oncolytic virus therapy in DIPG, and an important limitation is the difficulty of its administration. Currently, oncolytic viruses must be administered stereotactically to tumors through invasive surgery. Maciej et al. utilized mesenchymal stem cells as carriers to carry the modified oncolytic virus CRAd.S.pK7 and administered it through subcutaneous injection or intraperitoneal injection. The homing capacity of mesenchymal stem cells made it possible to cross the blood–brain barrier and enter the tumor. In addition, the stem cell-mediated delivery of CRAd.S.pK7 ensured virus dissemination throughout the tumor, delayed virus clearance by the host immune system and reduced neuroinflammatory responses, thereby providing neuroprotection to normal brain tissue [44,62,63]. In animal orthotopic model experiments, the results showed that the injection of CRAd.S.pK7-loaded mesenchymal stem cells after radiotherapy could lead to better survival in animals than either treatment alone [44].

### 3.4. ICI Therapy

Immune checkpoints are essentially costimulatory molecules that affect T-cell responses. They regulate or inhibit T-cell function by binding to corresponding ligands or receptors on the surface of T cells [64]. Inhibitory immune checkpoints, such as PD-L1, PD-L2 and CTLA-4, are highly expressed in many tumors, and they impair the function of cytotoxic T cells in the tumor [65,66,67]. Currently, the FDA has approved the ICIs CTLA-4 and PD-1, as well as their ligands PD-L1 and PD-L2, for specific tumor treatment [68,69]. Inhibitors targeting more immune checkpoints, such as LAG3, TIGIT, TIM3, B7H3, CD39, CD73, adenosine A2A receptor and CD47, are still under investigation.

Great progress has been made in ICI therapy in recent years. Previous studies have shown that ICIs can exert therapeutic effects on various tumors, including malignant melanoma, non-small cell lung cancer, renal cell carcinoma, gastric cancer and esophageal cancer [70,71,72,73,74,75,76]. However, the studies of ICI therapy in brain tumors are few, and the efficiency is not significant [77,78,79,80,81].

For clinical research on the role of ICIs in DIPG, the current research data are limited and the results are not consistent. In a retrospective study involving two DIPG patients who received both PD-1 inhibitor and CTLA-4 inhibitor therapy for a short duration before the discontinuation of ICI used to curtail disease progression, the results showed that there was no obvious drug toxicity in patients, but the survival time was also not significantly prolonged [82]. In a retrospective study of 31 recurrent DIPG patients, 8 patients received both reradiation and PD-1 inhibitor combination therapy, and their survival time was not significantly different but slightly prolonged compared with those in the control group, and none developed acute or chronic toxicity [83]. However, in another clinical trial that enrolled five DIPG children with a median age of 3.5 years, the PD-1 inhibitor Pembrolizumab was intravenously administered at 2 mg/kg every three weeks, and the patients experienced clinical or radiographic deterioration after 1 or 2 doses of treatment. Their PFS was only 1.02 months (range, 0.5–1.7), and OS from initial treatment was 1.7 months (range, 0.5–6.2), which was much shorter than in previous clinical studies. All patients clinically and/or radiographically worsened after one (*n* = 1) or two (*n* = 4) doses of pembrolizumab. In addition, 40% of enrolled patients (*n* = 2) had grade 3 or 4 treatment-related adverse events, including fatigue (*n* = 2) and new or increased grade neurologic symptoms [84].

### 3.5. Combination Therapy

Overall, the intersection and combination of immunotherapy with other treatments have increased in clinical trials of DIPG. Since the standard treatment for DIPG is radiotherapy, there is often a combination of radiation therapy and immunotherapy. The use of ^131^I-omburtamab to treat DIPG is one kind of radioimmunotherapy that is currently involved in a clinical trial for DIPG (clinicaltrials.gov: NCT05063357). Omburtamab is a monoclonal antibody that targets B7-H3 on the surface of DIPG, and ^131^I labeling can kill tumor cells through the effect of internal irradiation. In addition, there are many other combination therapies, including combinations of multiple immunotherapies or combinations of immunotherapy and chemotherapy, which are also involved in clinical trials (clinicaltrials.gov: NCT04943848, NCT02960230).

### 3.6. Summary of Immunotherapy Research on DIPG

Above all, the current immunotherapy research on DIPG mainly includes five parts: adaptive cell transfer therapy, oncolytic virus therapy, vaccine therapy, ICI therapy and combined therapy. From the results of clinical trials, the first three are the most promising new treatment options for DIPG.

Among them, GD2 CAR-T showed a good effect on prolonging the OS and obvious neuroinflammatory response [24]. How to weaken its neuroinflammatory response and intensify its anti-tumor effect may be the focus of future research and development.

For oncolytic viruses, DNX-2401 experiments have shown that it has good safety and efficacy [53,56,57]. However, studies have also shown that its changes in immunity are very short-lived and that it will increase adverse effects when combined with radiotherapy. In addition, Oncolytic viruses still require in situ inoculation, which limits their clinical use.

For vaccines, the H3.3K27M peptide vaccine significantly prolongs the OS of patients with immune response, but its response rate is only 39% [35]. How to improve the response rate of vaccines may be an important issue in future research.

In terms of ICI therapy, it is not a reasonable strategy for DIPG as immune characterization studies have shown that the expression of immune checkpoints is not significantly elevated in DIPG. ICI treatment should be considered in combination with other therapies to improve the therapeutic effect.

## 4. Conclusions

The development of immunotherapy has brought new potential prospects for DIPG, which is unresectable and lacks effective treatment strategies. The existing research shows that the immune microenvironment of DIPG presents a state of cold or immunodeficiency status, specifically manifested as a lack of T lymphocyte infiltration, noninflammatory performance of macrophages, low levels of regulating cytokine secretion and low levels of immune checkpoint expression [17]. Some clinical trials have shown significant effects on prolonging the OS of DIPG patients, which could supplement T cells directly or stimulate the adaptive immune system to infiltrate specific cytotoxic cells into tumors, involving adaptive T-cell transfer therapy, vaccine therapy and oncolytic virus therapy [24,35,56]. Although studies on the immune microenvironment are still insufficient and most clinical trials involving DIPG are still in phase 1 or phase 2, the results presented in this research provide a key background for future DIPG research.

## Figures and Tables

**Figure 1 cancers-15-00602-f001:**
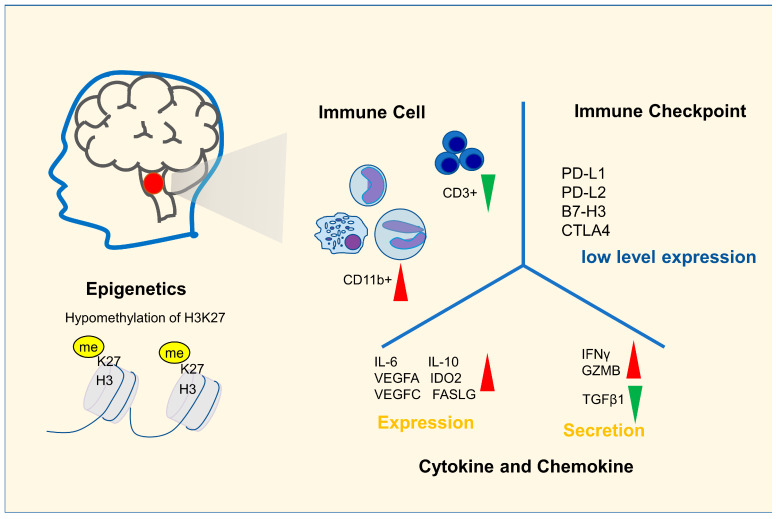
The epigenetic and immune microenvironment characteristics of DIPG. Most DIPGs have hypomethylation of H3K27. Compared with other high-grade gliomas, DIPG has lower CD3^+^T cell infiltration and a higher proportion of CD11b^+^ cell infiltration in its immune microenvironment. There is little expression of inhibitory immune checkpoints in DIPG, the expression of immunosuppressive cytokines is relatively low and the expression of proinflammatory factors is relatively high compared with high grade glioma.

**Figure 2 cancers-15-00602-f002:**
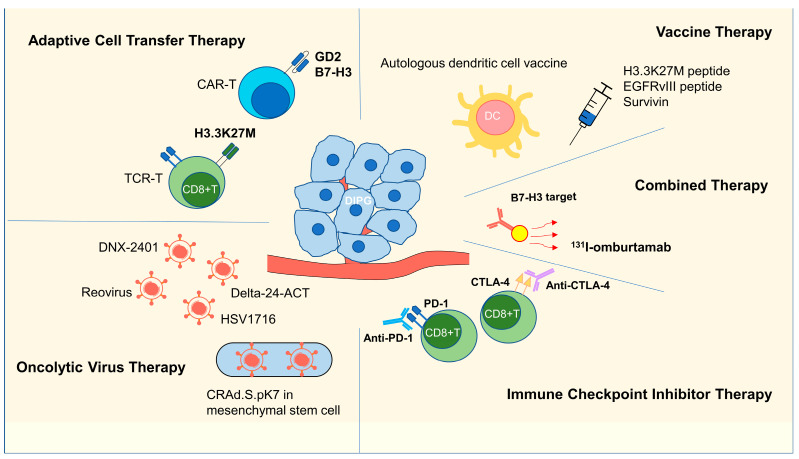
Immunotherapy strategies for DIPG. CAR T cells and TCR T cells are designed to target DIPG cells. Oncolytic viruses are designed to infect DIPG cells, kill them directly or attenuate their progression. A vaccine is injected to activate the adaptive immune system, especially CD8+ cytotoxic T cells. Anti-PD-1 and anti-CTLA-4 antibodies bind to PD-1 and CTLA-4, impeding the inactivation of CD8+ T cells, respectively. ^131^I-omburtamab can kill DIPG cells by targeting B7-H3 and internal irradiation. CAR T, chimeric antigen receptor T cells; TCR T, T-cell receptor-gene engineered T cells; CTLA-4, cytotoxic T-lymphocyte-associated protein 4; PD-1, programmed cell death protein 1; DC, dendritic cell.

**Table 1 cancers-15-00602-t001:** The clinical trials of immunotherapies for DIPG.

Classification	NCT Number	Title	Status	Interventions	Age (Years)	Phases	Enrollment	Combination with Other Therapy
Adoptive Transfer Cell Therapy	NCT04196413	GD2 CAR T Cells in Diffuse Intrinsic Pontine Gliomas (DIPG) and Spinal Diffuse Midline Glioma (DMG)	Recruiting	GD2 CAR T cells + Fludarabine +Cyclophosphamide	2–30	Phase 1	54	Combination with Chemotherapy
	NCT04099797	C7R-GD2.CAR T Cells for Patients with GD2-expressing Brain Tumors (GAIL-B)	Recruiting	(C7R)-GD2.CART cells + Cyclophosphamide+ Fludarabine	1–21	Phase 1	34	Combination with Chemotherapy
	NCT05298995	GD2-CAR T Cells for Pediatric Brain Tumors	Not yet recruiting	GD2-CART01 (iC9-GD2-CAR T cells)	0.5–30	Phase 1	54	No
	NCT04185038	Study of B7-H3-Specific CAR T-Cell Locoregional Immunotherapy for Diffuse Intrinsic Pontine Glioma/Diffuse Midline Glioma and Recurrent or Refractory Pediatric Central Nervous System Tumors	Recruiting	SCRI-CARB7H3(s); B7H3-specific chimeric antigen receptor (CAR) T cells	1–26	Phase 1	90	No
Vaccine	NCT02960230	H3.3K27M Peptide Vaccine With Nivolumab for Children With Newly Diagnosed DIPG and Other Gliomas	Active, not recruiting	K27M peptide+ Nivolumab	3–21	Phase 1|Phase 2	50	Combination with an Anti-PD-1 mAb
	NCT04749641	Neoantigen Vaccine Therapy Against H3.3-K27M Diffuse Intrinsic Pontine Glioma	Recruiting	Histone H3.3-K27M Neoantigen Vaccine	5-	Phase 1	30	No
	NCT04808245	A MultIceNTER Phase I Peptide VaCcine Trial for the Treatment of H3-Mutated Gliomas	Not yet recruiting	Tecentriq +H3K27M peptide vaccine+ Imiquimod (5%)	18-	Phase 1	15	Combination with an Anti-PD-L1 mAb
	NCT01058850	Phase I Rindopepimut After Conventional Radiation in Children w/Diffuse Intrinsic Pontine Gliomas	Terminated	Rindopepimut	3–18	Phase 1	3	No
	NCT04978727	A Pilot Study of SurVaxM in Children Progressive or Relapsed Medulloblastoma, High Grade Glioma, Ependymoma and Newly Diagnosed Diffuse Intrinsic Pontine Glioma	Recruiting	SurVaxM	1–21	Phase 1	35	Part of patients combination with radiotherapy
	NCT04943848	rHSC-DIPGVax Plus Checkpoint Blockade for the Treatment of Newly Diagnosed DIPG and DMG	Recruiting	rHSC-DIPGVax+ Balstilimab+ Zalifrelimab	1–18	Phase 1	36	Combination an Anti-PD-1 mAb and an Anti-CTLA4 mAb
	NCT02750891	A Study of DSP-7888 in Pediatric Patients with Relapsed or Refractory High-grade Gliomas	Completed	DSP-7888	0–19	Phase 1|Phase 2	18	No
	NCT02840123	Safety Study of DIPG Treatment with Autologous Dendritic Cells Pulsed with Lysates from Allogenic Tumor Lines	Unknown status	Autologous dendritic cells	3–18	Phase 1	10	No
	NCT03914768	Immune Modulatory DC Vaccine Against Brain Tumor	Enrolling by invitation	Immunomodulatory DC vaccine to target DIPG and GBM	1–75	Phase 1	10	No
	NCT04911621	Adjuvant Dendritic Cell Immunotherapy for Pediatric Patients With High-grade Glioma or Diffuse Intrinsic Pontine Glioma	Recruiting	Dendritic cell vaccination + temozolomide-based chemoradiation+- conventional next-line treatment	1–17	Phase 1|Phase 2	10	Combination with Temozolomide-based Chemoradiation
	NCT03396575	Brain Stem Gliomas Treated with Adoptive Cellular Therapy During Focal Radiotherapy Recovery Alone or with Dose-intensified Temozolomide (Phase I)	Recruiting	TTRNA-DC vaccines with GM-CSF/TTRNA-xALT/cyclophosphamide + fludarabine lymphodepletive conditioning/dose-intensified TMZ/Td vaccine/autologous hematopoietic stem cells (HSCs)	3–30	Phase 1	21	Combination with GM-CSF or Chemotherapy or Autologous Hematopoietic Stem Cells
	NCT02750891	A Study of DSP-7888 in Pediatric Patients with Relapsed or Refractory High-grade Gliomas	Completed	DSP-7888	0–19	Phase 1|Phase 2	18	No
Oncolytic Virus Therapy	NCT03330197	A Study of Ad-RTS-hIL-12 + Veledimex in Pediatric Subjects with Brain Tumors Including DIPG	Terminated	Ad-RTS-hIL-12+ Oral Veledimex	0–21	Phase 1|Phase 2	6	No
	NCT03178032	Oncolytic Adenovirus, DNX-2401, for Naive Diffuse Intrinsic Pontine Gliomas	Active, not recruiting	DNX-2401	1–18	Phase 1	12	No
	NCT02444546	Wild-Type Reovirus in Combination with Sargramostim in Treating Younger Patients With High-Grade Relapsed or Refractory Brain Tumors	Active, not recruiting	Sargramostim+ Wild-type Reovirus	10–21	Phase 1	6	Combination with Sargramostim
	NCT04758533	Clinical Trial to Assess the Safety and Efficacy of AloCELYVIR with Newly Diagnosed Diffuse Intrinsic Pontine Glioma (DIPG) in Combination with Radiotherapy or Medulloblastoma in Monotherapy	Recruiting	AloCELYVIR	1–21	Phase 1|Phase 2	12	No
	NCT05096481	PEP-CMV Vaccine Targeting CMV Antigen to Treat Newly Diagnosed Pediatric HGG and DIPG and Recurrent Medulloblastoma	Not yet recruiting	PEP-CMV+ Temozolomide+ Tetanus Diphtheria Vaccine	3–25	Phase 2	120	Combination with Temozolomide
Immune Checkpoint Inhibitor Therapy	NCT01952769	Anti PD1 Antibody in Diffuse Intrinsic Pontine Glioma	Unknown status	MDV9300	3–21	Phase 1|Phase 2	50	No
	NCT02359565	Pembrolizumab in Treating Younger Patients With Recurrent, Progressive, or Refractory High-grade Gliomas, Diffuse Intrinsic Pontine Gliomas, Hypermutated Brain Tumors, Ependymoma or Medulloblastoma	Recruiting	Pembrolizumab	1–29	Phase 1	110	No
	NCT03690869	REGN2810 in Pediatric Patients with Relapsed, Refractory Solid, or Central Nervous System (CNS) Tumors and Safety and Efficacy of REGN2810 in Combination With Radiotherapy in Pediatric Patients With Newly Diagnosed or Recurrent Glioma	Recruiting	Cemiplimab	0–25	Phase 1|Phase 2	130	No
Other	NCT05063357	^131^I-omburtamab Delivered by Convection-Enhanced Delivery in Patients with Diffuse Intrinsic Pontine Glioma	Not yet recruiting	^131^I-Omburtamab	3–21	Phase 1	36	No
	NCT01502917	Convection-Enhanced Delivery of 124I-Omburtamab for Patients with Non-Progressive Diffuse Pontine Gliomas Previously Treated with External Beam Radiation Therapy	Completed	Radioactive iodine-labeled monoclonal antibody omburtamab+ external beam radiotherapy	2–21	Phase 1	50	Radiotherapy

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
