# Peer review of "Immune Microenvironment and Immunotherapies for Diffuse Intrinsic Pontine Glioma"

_cancers, 2023, doi:10.3390/cancers15030602_

Round 1

Reviewer 1 Report (Previous Reviewer 3)

The authors have amended the text in response to my comments and I think it now is significantly easier to read and has increased clarity.

Reviewer 2 Report (Previous Reviewer 1)

The authors have thoroughly revised the manuscript and have included all suggestions of the reviewer. 

This manuscript is a resubmission of an earlier submission. The following is a list of the peer review reports and author responses from that submission.

Round 1

Reviewer 1 Report

This is a review paper on the immune characteristics of pontine / midline glioma and a review of the challenges and opportunities of immunotherapies in these tumors.

First of all and importantly to solve, "DIPG" is not a histological, but more morphological diagnosis, and H3K27 midline glioma is not the histological counterpart to "DIDG". This should  be corrected and clearly indicated in the manuscript. The correction is also important as the authors also discuss trials in other midline entities as H3K27. So, the title would better include something like "midline glioma" or "paediatric midline glioma".

Besides this, the paper is a complete review of many immunotherapies that have been investigated in midline gliomas. Tables are also complete. It is beyond  the scope of my expertise to check the completeness and every detail of trials cited in the paper, but it seems that authors have done  a very good job to compile every data point that os available. 

The completeness of the review leads to a number of studies discussed in a row. In my opinion, authors should add another chapter at the end of the article where they discuss the advantages and disadvantages of the approaches, weigh the therapeutic means and present a clear concept how to go on in this research. This is of course somehow subjective, but would the review make more interesting to read also for a broad readership which is not interested in any detail.

Citation 9 claims to cite the 5th version of the WHO classification, but this is not correct. Please cite the correct paper and check all citations for their suitability.

The English language is quite well to understand, but has many grammar errors that have to be corrected. This has to be significantly improved. I would suggest a thorough review and rewording by a native speaker.

Author Response

Answer to Reviewer #1

Reviewer #1: This is a review paper on the immune characteristics of pontine / midline glioma and a review of the challenges and opportunities of immunotherapies in these tumors.

Comment1: First of all and importantly to solve, "DIPG" is not a histological, but more morphological diagnosis, and H3K27 midline glioma is not the histological counterpart to "DIDG". This should be corrected and clearly indicated in the manuscript. The correction is also important as the authors also discuss trials in other midline entities as H3K27. So, the title would better include something like "midline glioma" or "paediatric midline glioma".

Answer:

Thanks for your comment. The question of the relationship between DIPG and DMG is important and has been revised in the manuscript. In fact, as you said, DIPG is a clinical diagnosis. If there is H3K27 alteration, we diagnose it as “diffuse midline glioma, H3 K27-altered (DMG)”, otherwise we diagnose it as “pediatric diffuse high-grade glioma, H3 wild-type and IDH wild-type”. Therefore, the relationship between DIPG and DMG is not an inclusion relationship, but an intersection relationship. We explain this situation further in the manuscript and marked it in red.

Also, we appreciate your suggestion to modify the title to include midline glioma or pediatric midline glioma. After careful consideration, we hope to retain the original title. This is mainly due to the fact that our core concerns are still the immune characteristics and the immunotherapy research of DIPG. In the section on the tumor microenvironment, many studies were conducted in children with brainstem tumors, which did not differentiate whether the diagnosis of DMG was met. And although our article involves multiple experimental studies of DMG, we still focus on the results of patients with DIPG in our research. Therefore, in general, we are still a research overview on DIPG.

Comment2: Besides this, the paper is a complete review of many immunotherapies that have been investigated in midline gliomas. Tables are also complete. It is beyond the scope of my expertise to check the completeness and every detail of trials cited in the paper, but it seems that authors have done a very good job to compile every data point that os available. 

Answer:

Thanks for your comment. In fact, we have strictly checked the screening of clinical trials and the production of forms. The clinical trials shown in Table 1 include DIPG and DMG mainly because the two diagnoses overlap. Although some clinical trials did not mention DIPG, but in fact they might enroll H3K27 altered DIPG patients.

Comment3: The completeness of the review leads to a number of studies discussed in a row. In my opinion, authors should add another chapter at the end of the article where they discuss the advantages and disadvantages of the approaches, weigh the therapeutic means and present a clear concept how to go on in this research. This is of course somehow subjective, but would the review make more interesting to read also for a broad readership which is not interested in any detail.

Answer:

Thank you for your comment. We think your comment is very meaningful. In this revision, we have added section 3.6 based on your suggestion, combining the discovery of the immune microenvironment and the data of immunotherapy to elaborate the feasibility and possible future development of different immunotherapy methods in DIPG.

Comment4: Citation 9 claims to cite the 5th version of the WHO classification, but this is not correct. Please cite the correct paper and check all citations for their suitability.

Answer:

Thanks for your comment. We are very sorry that our citation 9 was inaccurate. We have revised this citation and checked other citations for accuracy.

Comment5: The English language is quite well to understand, but has many grammar errors that have to be corrected. This has to be significantly improved. I would suggest a thorough review and rewording by a native speaker.

Answer:

Thanks for your comment. We are very sorry for the grammar errors in the manuscript. Based on your suggestion, we commissioned AJE to carry out a comprehensive language revision of this manuscript.

Reviewer 2 Report

Chen et al. provide a comprehensive review of immune microenvironment and immunotherapy for diffuse midline gliomas. The following points should be addressed to further improve the paper.

Main points:

1. English language editing is advised.

2. In the abstract, while it is true that DIPGs are not amenable to radical surgery and are insensitive to chemotherapies, the claim of rapid growth (line 17) is controversial. In fact, many DIPG cell lines and xenografts are known to be slow growing compared to GBM (PMID 26115193, 28450157).

3. Follow up of the important GD2 CART therapy trial for DMG has been presented at AACR2021 and should be referenced. https://aacrjournals.org/cancerres/article/81/13_Supplement/CT031/669739

Minor point:

Line 41 "amendable" should be "amenable"

Author Response

 Answer to Reviewer #2

Chen et al. provide a comprehensive review of immune microenvironment and immunotherapy for diffuse midline gliomas. The following points should be addressed to further improve the paper.

Main points:

Comment1: English language editing is advised.

Answer:

Thanks for your comment. We are very sorry for the grammar errors and spelling errors in the manuscript. Based on your suggestion, we commissioned AJE to carry out a comprehensive language revision of this manuscript.

Comment2: In the abstract, while it is true that DIPGs are not amenable to radical surgery and are insensitive to chemotherapies, the claim of rapid growth (line 17) is controversial. In fact, many DIPG cell lines and xenografts are known to be slow growing compared to GBM (PMID 26115193, 28450157).

Answer:

Thanks for your comment. Your opinion is quite correct, there is the semantic confusion caused by our expression. In fact, as you said, the growth of the DIPG cell lines and tumor xenografts are significantly slower compared to GBM. What we want to express in the manuscript is that DIPG has a very short survival period from diagnosis to patient death, and the disease progresses rapidly. Now we have revised it as “rapid progression” in the manuscript.

Comment3: Follow up of the important GD2 CART therapy trial for DMG has been presented at AACR2021 and should be referenced. https://aacrjournals.org/cancerres/article/81/13_Supplement/CT031/669739

Answer:

Thanks for your comment. The research you mentioned is indeed very important, it strongly shows that GD2 CAR-T therapy is effective for DIPG patients. Its abstract was published on Cancer Research in 2021 as you quoted, and its main text was published on Nature in 2022, which is the citation 20 in our original manuscript. In the revision of the manuscript, we have also cited the abstract you mentioned in the text.

Minor point:

Comment4: Line 41 "amendable" should be "amenable"

Answer:

Thanks for your comment. I am very sorry for such a spelling error. We have corrected it in the manuscript and checked the spelling of other words.

Reviewer 3 Report

This is a comprehensive review of the immune environment in DIPG/DMG. The manuscript needs extensive reviewing for grammar and phrasing as some of the sentences are incomplete or unclear. The manuscript provides a good summary if of current trials in DIPG immune therapy approaches but needs revising for written style and the comments I outline below.

My specific comments are:

- The authors need to be specific about what they mean by DIPG/DMG. DIPG is not an entity listed in the WHO CNS5 classification and is a clinical diagnosis based on symptoms and signs and typical imaging. The actual underlying histological diagnosis is DMG. It would be useful to provide clarity on this in the manuscript.

- Line 10-11 : the sentence is incomplete and I am unsure what it is intended to mean.

- Line 23: Stereotactic spelt incorrectly.

- Sections 2.1 and 2.2 are very similar and overlap in places. I would merge these two sections for clarity.  

- It would be useful to reflect on the strength of some of the evidence presented, particularly in the immune environment/immune infiltration sections. For example, the study quoted as reference 11 using cibersort computational techniques has no real additional validation and uses blood (LM22) as a matrix for the cibersort analysis rather than tumour-derived immune cell signatures. It would be useful for the authors to provide their own view on the strength of this evidence and whether we can be confident that there really is an immune-neutral environment in DIPG on the basis of this?

- The implications of the findings on lines 124-127 need to be described in more detail.

Author Response

Answer to Reviewer #3

This is a comprehensive review of the immune environment in DIPG/DMG. The manuscript needs extensive reviewing for grammar and phrasing as some of the sentences are incomplete or unclear. The manuscript provides a good summary if of current trials in DIPG immune therapy approaches but needs revising for written style and the comments I outline below.

My specific comments are:

Comment1: The authors need to be specific about what they mean by DIPG/DMG. DIPG is not an entity listed in the WHO CNS5 classification and is a clinical diagnosis based on symptoms and signs and typical imaging. The actual underlying histological diagnosis is DMG. It would be useful to provide clarity on this in the manuscript.

Answer:

Thanks for your comment. The question of the relationship between DIPG and DMG is important and has been revised in recognition of the lack of clarity in the manuscript. In fact, as you said, DIPG is a clinical diagnosis. If there is H3K27 alteration, we diagnose it as “diffuse midline glioma, H3 K27-altered (DMG)”, otherwise we diagnose it as “pediatric diffuse high-grade glioma, H3 wild-type and IDH wild-type”. Therefore, the relationship between DIPG and DMG is not an inclusion relationship, but an intersection relationship. We explain this situation further in the manuscript and marked it in red.

Comment2: Line 10-11 : the sentence is incomplete and I am unsure what it is intended to mean.

Answer:

Thank you for your comment. I am very sorry for the language confusion. We have revised the phrased way of these two sentences in the new revision.

Comment3: Line 23: Stereotactic spelt incorrectly.

Answer:Thank you for your comment. We have carefully considered your suggestion and rewritten stereotaxic into stereotactic according to your suggestion.

Interestingly, after consulting the literature, we found that the two spellings of this word both coexist, and there has been an interesting debate about these two spellings in history (1,2). Nowadays, although these two spellings are still used, the “stereotactic” is significantly more used than “stereotaxic”. Therefore, we have modified the spelling of the word.

Comment4: Sections 2.1 and 2.2 are very similar and overlap in places. I would merge these two sections for clarity.  

Answer:

Thank you for your suggestion. In fact, as you said, there is a considerable proportion of overlap between these two chapters. This is because most research on the immune microenvironment in DIPG always explore the immune cell infiltration and the immune molecular expression at the same time. In the new revision, we have removed the subtitle according to your suggestion, and synthesized the two parts into a unified part, namely the immune characteristics of DIPG.

Comment5: It would be useful to reflect on the strength of some of the evidence presented, particularly in the immune environment/immune infiltration sections. For example, the study quoted as reference 11 using cibersort computational techniques has no real additional validation and uses blood (LM22) as a matrix for the cibersort analysis rather than tumour-derived immune cell signatures. It would be useful for the authors to provide their own view on the strength of this evidence and whether we can be confident that there really is an immune-neutral environment in DIPG on the basis of this?

Answer:

Thank you for your comment. We think that your opinion is helpful. In the previous manuscript, we mainly focused on the consistency of the findings while ignoring the strength of the evidence presented. In this new revision, we have carefully checked the strength of evidence presented and added details on the research methods for readers to judge the credibility of the evidence.

Comment6: The implications of the findings on lines 124-127 need to be described in more detail.

Answer:

Thank you for your suggestion. We have expanded this paragraph to include more details.

Reference

  1. Gildenberg PL. Stereotactic versus stereotaxic. Neurosurgery 1993;32(6):965-6.
  2. Suss RA. Stereotactic versus stereotaxic. Neurosurgery 1993;33(6):1114.